# Diabetic Kidney Disease versus Primary Glomerular Disease: A Propensity Score-Matched Analysis of Association between Ambulatory Blood-Pressure Monitoring and Target-Organ Damage

**DOI:** 10.3390/jcm12010167

**Published:** 2022-12-25

**Authors:** Tiantian Yu, Shicong Song, Xiaoqiu Chen, Tanqi Lou, Jun Zhang, Hui Peng, Man Li, Cheng Wang

**Affiliations:** 1Division of Nephrology, Department of Medicine, The Fifth Affiliated Hospital Sun Yat-sen University, 52 Meihua East Road, Zhuhai 519000, China; 2Guangdong Provincial Key Laboratory of Biomedical Imaging, The Fifth Affiliated Hospital Sun Yat-sen University, 52 Meihua East Road, Zhuhai 519000, China; 3Division of Nephrology, Department of Medicine, Third Affiliated Hospital of Sun Yat-sen University, Guangzhou 510630, China

**Keywords:** diabetic kidney disease, primary glomerular disease, ambulatory blood-pressure monitoring, target-organ damage, propensity-score matching

## Abstract

Diabetic kidney disease (DKD) and primary glomerular disease (PGD) are the main causes of chronic kidney disease (CKD) and end-stage renal disease (ESRD). This study was conducted to compare the characteristics of ambulatory blood-pressure monitoring (ABPM) and its relationship with target-organ damage (TOD) in patients with DKD and PGD matched by propensity score. The assessment of TOD included macroalbuminuria, left ventricular hypertrophy (LVH) and macrovascular disease. Propensity-score weighting (PSW) was used in stratified analysis. Results: Patients with DKD had a higher prevalence of abnormal blood-pressure patterns such as reversed dipper pattern, nocturnal hypertension, and sustained hypertension and had a higher prevalence of TOD than did patients with PGD. Logistic regression indicated that patients with DKD were more related to TOD than to PGD. The stratified analysis indicated that DKD patients with white-coat hypertension, masked hypertension and sustained hypertension had closer relationships with TOD compared with PGD patients. Conclusion: Patients with type 2 diabetic kidney disease had more abnormal blood-pressure patterns and were more closely related to target organ damage than were patients with primary glomerular disease.

## 1. Background

Type 2 diabetic mellitus (DM) is an important worldwide public health problem and affects more than 20 million people around the world [1,2]. Currently diabetic kidney disease (DKD) as a major microvascular complication of DM is responsible for up to 40% of all causes of chronic kidney disease (CKD) and end-stage renal disease (ESRD) [3,4,5,6,7]. China has also been suffering from an increasing prevalence of DKD which may lead to high cardiovascular risks and mortality [8,9,10,11,12,13]. DKD and primary glomerular disease (PGD) are the two most common causes of CKD and ESRD all over China and around the world [8,9].

Hypertension is an independent risk factor for the development of DKD [14,15,16]. Controlled blood pressure (BP) has been proved to be effective in postponing the progression of renal failure and in reducing overall mortality [16]. Accurate measurement of BP and early detection of hypertension are essential to assess cardiovascular risks. Considering the limitations of clinical blood-pressure measurement, ambulatory blood-pressure monitoring (ABPM) has been paid more and more attention. ABPM can not only monitor blood pressure throughout the day and find blood-pressure variation, but also detect important abnormal blood-pressure patterns [17]. 

Some previous studies have shown the superiority of ABPM over clinical BP measurements, and have suggested that ABPM is better in predicting cardiovascular outcomes in the general population and in patients with hypertension or CKD [18,19,20,21,22,23,24]. However, few studies have described ABPM characteristics in patients with DKD, and the research focusing on comparation between patients with DKD and PGD is even less. Most research has focused on patients with diabetes only [25,26,27]. Moreover, most of the subjects of previous studies were patients with type 1 diabetes.

Therefore, we decided to conduct our study through matching DKD and PGD patients by propensity score to investigate and compare the characteristics of ABPM and the association between ABPM and target-organ damage (TOD).

## 2. Methods

### Participants

This work was supported by the Five-five Project of the Fifth Affiliated Hospital of Sun Yat-sen University. The study protocol was approved by the ethics committee of our hospital (K14-1) and adhered to the Declaration of Helsinki. Written informed consent was given by all participants.

Type 2 DKD patients: type 2 diabetic patients aging from 15 to 75 with persistent presence of elevated urinary albumin excretion, decreased eGFR, or other manifestations of kidney damage without signs or symptoms of other primary or secondary kidney damage according to the 2020 American Diabetes Association were enrolled.

Two of three specimens of urinary albumin to creatinine ratio (UACR) collected within a 3 to 6 month period should be more than 30 mg/gCr excluding exercise within 24 h, infection, fever, congestive heart failure, marked hyperglycemia and menstruation [24]. The eGFR value was calculated from serum creatinine using the 2009 Chronic Kidney Disease Epidemiology Collaboration (CKD-EPI) equation. Decreased eGFR was defined as eGFR persistently less than 60 mL/min per 1.73 m^2^. 

Patients with PGD: patients with the signs and symptoms of kidney damage or decreased renal function (eGFR <60 mL/min per 1.73 m^2^) for more than 3 months excluding other secondary factors.

The exclusion criteria were as follows: maintenance dialysis or history of kidney transplantation; pregnancy; acute changes in eGFR >30% in the previous three months; atrial fibrillation; undergoing treatment with corticosteroids or hormones; night work or shift-work employment; intolerance to ABPM or invalid ABPM data; inability to communicate and comply with all of the study requirements.

Finally, 501 type 2 DKD patients and 2272 PGD patients were enrolled in this cross-sectional study and they were matched by the propensity score of age, sex and eGFR in a ratio of one to one. Therefore, 501 type 2 DKD patients and 501 PGD patients were finally enrolled.

## 3. Blood-Pressure Measurements

ABPM was performed via calibrated devices in our clinical centers, and programmed at 15 min intervals during the daytime and 30 min intervals at night using an appropriate cuff placed on the nondominant arm [28,29]. Day and night periods were defined according to sleeping and waking times reported by the patient. 

## 4. Target-Organ Damage 

Macroalbuminuria was defined as UACR ≥300 mg/gCr. Echocardiography was performed by two experienced cardiologists according to the recommendations of the American Society of Echocardiography and the European Association of Cardiovascular Imaging. Linear measurements of the end-diastolic interventricular septal-wall thickness (IVST), left ventricular end-diastolic diameter (LVEDD), and end-diastolic posterior wall thickness (PWT) were assessed using M-mode tracings using 2-dimensional echocardiography. Left ventricular mass (LVM) was calculated as LVM(g) = 0.8*{1.04*[(LVEDD + IVST + PWT)^3^ − LVEDD^3^]} + 0.6. Left ventricular hypertrophy (LVH) was defined through the left ventricular mass index (LVMI) according to recent guidelines, with LVM normalized to body surface area, as greater than 115 g/m^2^ in men and greater than 95 g/m^2^ in women [30,31]. Patients with clinical evidence of carotid intima–media thickness > 0.9 mm or carotid plaque, lower limb arteriosclerosis, coronary atherosclerosis, myocardial infarction or stroke were diagnosed with macrovascular diseases [32]. The methods of carotid intima–media thickness (CIMT) measurement was described in previous studies [33,34,35]. Bilateral lower limb arteries were examined with vascular ultrasound, and cerebrovascular disease was examined through brain magnetic-resonance imaging or computed tomography. Myocardial infarction was diagnosed through either a combination of electrocardiography and clinical syndromes or prior coronary angioplasty. 

## 5. Data Collection

Basic sociodemographic and clinical characteristics were collected. Medical history and current therapy were obtained from clinical records. A fasting blood sample was collected to measure hemoglobin, albumin, calcium, phosphorus, intact parathyroid hormone (iPTH), serum fasting glucose, glycosylated hemoglobin, cholesterol, triglycerides, high-density lipoprotein cholesterol (HDL-C), low-density lipoprotein-cholesterol (LDL-C), uric acid (UA), and serum creatinine (Scr), which were measured using a 7180 Biochemistry Auto-analyzer (Hitachi, Tokyo, Japan) in the central laboratory. We collected urine samples from 7 a.m. to 7 a.m. the next day to detect the extent of proteinuria over 24 h. These patients were asked to void their bladders before and after the urine collection. Proteinuria was measured by immunoturbidimetry.

## 6. Definitions

Nocturnal hypertension was defined as the average of night-time BP values at least 120/70 mmHg. According to BP at night, patients can be divided into four groups as extreme dipper, normal dipper, non-dipper and reversed dipper pattern. The difference of daytime and night-time systolic blood pressure (SBP) versus the value of daytime SBP can be calculated as a dipping rate. Extreme dipper pattern is defined as a dipping rate > 20%, and when the dipping rate is between 10% and 20%, normal dipper pattern is defined. Non-dipper pattern is called when the dipping rate is between 0 and 10%, and reversed dipper pattern is defined when the dipping rate is <0 [36,37].

Clinical hypertension is defined as clinical BP values at least 140/90 mmHg, or current use of antihypertensive medication. Further, 24-h ABPM hypertension was defined as average BP values of at least 130/80 mmHg. Combining measurements of clinical BP and ABPM, patients can also be divided into four different groups. Normotension was defined as clinical BP less than 140/90 mmHg and ambulatory BP less than 130/80 mmHg. White-coat hypertension (WCH) was defined as clinical BP at least 140/90 mmHg but ambulatory BP less than 130/80 mmHg. Masked hypertension (MH) was defined as clinical BP less than 140/90 mmHg but ambulatory BP at least 130/80 mmHg. Sustained hypertension was defined as clinical BP at least 140/90 mmHg and ambulatory BP at least 130/80 mmHg [36,37].

## 7. Statistical Analysis 

We matched patients with DKD with patients with PGD through propensity scores matching (PSM) with a one-to-one nearest neighbor caliper width of 0.01 (maximum allowable difference in propensity scores). Propensity score was calculated using a logistic regression model to estimate the probability of the disease assignment on the basis of variables such as age, sex and eGFR. Descriptive statistics are presented as mean ± standard deviation (SD) for continuous variables and median and interquartile range for nonparametric variables. Frequency and percentage were used for categorical variables. Log transformation for proteinuria and the eGFR in regression analyses were used because of the skewed distribution. Comparisons of continuous variables between groups were evaluated by the Student’s *t*-test, analysis of variance (ANOVA), or nonparametric test. Differences among categorical variables were analyzed using the chi-squared test or the two-tailed Fisher’s exact test. *p*-values for multiple comparisons were corrected according to the Bonferroni method. Univariate and multivariate logistic regression analyses were used to explore factors associated with target-organ damage and the results were expressed in terms of odds ratio (OR) with 95% CI. After univariate analyses, variables with clinical relevance and statistical significance were selected for multiple logistic regression. In stratified analysis, propensity score calculated by all known correlated covariates except for variate DKD (versus PGD) was used for weighting to eliminate imbalance between groups. Statistical analyses were performed using IBM SPSS Statistics Version 25.0 (IBM Corp., Armonk, New York, USA) and R Version 3.6.1 (R Foundation for Statistical Computing, Vienna, Austria. URL: https://www.R-project.org/. accessed on 1 May 2022) and *p* values of less than 0.05 were considered statistically significant.

## 8. Results

### 8.1. Demographic and Clinical Characteristics

The average age of 501 type 2 DKD patients was 57.3 years, and 66.5% of patients were men. Mean eGFR was 17.3 mL/min per 1.73 m^2^. The average age of 501 PGD patients was 57.9 years, and 64.5% of patients were men. Mean eGFR was 24.0 mL/min per 1.73 m^2^. (Table 1). No statistical significance was found between DKD and PGD groups on these three variables which indicated a good balance. The distribution of propensity score during the matching methods is shown. (Figure 1). The median of the course of kidney disease was 12 (2–48) months in the two groups. The percentage of patients receiving antihypertensive therapy in the DKD group was 88.2% and it was 80.0% in the PGN group. A total of 42.3% of DKD patients used RAS blockers and that number was 33.9% in PGN patients. In the DKD group, 93.4% of the patients received hypoglycemic treatment and 42% of patients were treated with insulin.

Patients in the DKD group compared with the PGD group showed higher BMI, using of antihypertensive drugs, using of statin, serum phosphate, iPTH, HbA1c, serum fasting glucose, cholesterol, UACR and LVMI (*p* < 0.05). Patients in the DKD group showed lower hemoglobin, albumin and LVEF (*p* < 0.05) (Table 1). 

### 8.2. Prevalence of Blood-Pressure Pattern

Patients in the DKD group compared with patients in the PGD group showed higher clinical systolic blood pressure (SBP), 24 h average SBP, daytime SBP and night-time BP (*p* < 0.05) (Table 1). The prevalence of nocturnal hypertension in the DKD group was 91.6% which was higher than 83.8% in the PGD group (Table 1). 

There were 9 (1.8%) patients with extreme dipper pattern, 67 (13.4%) patients with normal dipper pattern, 245 (48.9%) patients with non-dipper pattern, and 180 (35.9%) patients with reversed dipper pattern in the DKD group. There were 6 (1.2%) patients with extreme dipper pattern, 113 (22.6%) patients with normal dipper pattern, 247 (49.3%) patients with non-dipper pattern, and 134 (26.7%) patients with reversed dipper pattern in the PGD group. Compared with the PGD group, the DKD group showed a higher prevalence of reversed dipper pattern and lower normal dipper pattern (*p* < 0.05).

There were 51 (10.2%) patients with normotension and 330 (65.9%) patients with sustained hypertension in the DKD group. Misclassification was detected in 24% of DKD patients: 38 (7.6%) patients with white-coat hypertension and 82 (16.4%) patients with masked hypertension. There were 97 (19.4%) patients with normotension and 258 (51.5%) patients with sustained hypertension in the PGD group. Misclassification was detected in 29% of PGD patients: 45 (9.0%) patients with white-coat hypertension and 101 (20.2%) patients with masked hypertension. Compared with the PGD group, the DKD group showed a higher prevalence of sustained hypertension and fewer patients with normal BP (*p* < 0.05). (Figure 2).

### 8.3. Prevalence of Target Organ Damage

The prevalence of macroalbuminuria, LVH and macrovascular diseases were 72.3%, 65.9% and 63.1% in the DKD group, respectively, and were 60.1%, 47.1% and 45.9% in the PGD group. The prevalence rates of macroalbuminuria, LVH and macrovascular disease were all significantly higher in the DKD group than in the PGD group (*p* < 0.05) (Table 2).

Age, sex, BMI, current smoking status, alcohol intake, antihypertensive drugs, statin, serum fasting glucose, triglyceride, cholesterol, HDL-c, LDL-c, hemoglobin, HbA1c, serum albumin, uric acid, serum calcium, serum phosphate, iPTH, and eGFR were used in logistic regression for each TOD. After univariate analyses, variables with clinical relevance and statistical significance (*p* < 0.05) were selected for multiple logistic regression. 

Variate DKD (versus PGD) was associated with TOD like macroalbuminuria (1.730 (1.328–2.255), *p* < 0.001), LVH (2.496 (1.771–3.426), *p* < 0.001) and macrovascular disease (2.139 (1.620–2.824), *p* < 0.001) in univariate logistic regressions as seen in Model 1 (Table 3). After adjusting other covariates, Variate DKD (versus PGD) was still independently associated with macroalbuminuria (1.707 (1.304–2.235), *p* < 0.001), LVH (2.267 (1.715–2.999), *p* < 0.001) and macrovascular disease (2.107 (1.602–2.771), *p* < 0.001) in multivariate logistic regressions as seen in Model 2 (Table 3). When DKD (versus PGD) and hypertension type (versus normotension) were put together into the multivariate logistic regression analysis, DKD (versus PGD) was still independently associated with TOD; meanwhile, compared with normotension, WCH, MH and sustained HBP were also independently associated with macroalbuminuria and LVH (Model 3) (Table 3). 

DKD (versus PGD) and Nocturnal hypertension (Yes/No) were also independently related to macroalbuminuria, LVH and macrovascular diseases in multivariable logistic regression (Table 4).

### 8.4. Stratified Analysis between Group and TOD in Different Hypertension Types

We divided all of our patients into four groups including a normotension group, white-coat hypertension group, masked hypertension group and sustained hypertension group. In each group, propensity score calculated by all known correlated covariates except for DKD (versus PGD) was used for weighting to eliminate the imbalance between groups. 

The associations between DKD (versus PGD) and TOD in four types of hypertension groups were different. In detail, DKD patients with WCH, MH and sustained HBP were more associated with LVH and macrovascular disease than were PGD patients (Figure 3). As for macroalbuminuria, DKD patients with sustained HBP, MH and controlled BP had a closer relationship with macroalbuminuria than did PGD patients. 

## 9. Discussion

We collected clinical data of 501 Chinese hospitalized type 2 DKD patients and 2272 PGD patients and enrolled 501 of the PGD patients who were matched by age, sex and eGFR through propensity score in our study. Compared with PGD patients, DKD patients had a higher prevalence of reversed dipper pattern, nocturnal hypertension, sustained hypertension and had a higher prevalence of TOD. Logistic regression indicated that patients with DKD were more related to TOD than were patients with PGD. The stratified analysis indicated that DKD patients with white-coat hypertension, masked hypertension and sustained hypertension had a closer relationship with TOD compared with PGD patients. These results indicated that under limited resources, we may put more attention on patients with DKD instead of those with PGD.

Recent research has shown that ambulatory blood-pressure monitoring has a closer relationship with cardiovascular risk than with clinical blood pressure in patients with hypertension. In addition, compared with clinical blood pressure, ambulatory blood pressure provided more specific and accurate information on renal and cardiovascular prognosis in patients with chronic kidney disease. However, the differences in characteristics of ambulatory blood-pressure monitoring between patients with diabetic kidney disease and primary glomerular disease still remains unclear. As two of the most common causes of CKD and ESRD, these patients accounted for almost 60% to 70%, and DKD has surpassed PGD to be the leading cause of CKD and ESRD in recent years in China. Our present study focused on the differences between type 2 DKD patients and PGD patients and revealed the importance of ABPM in DKD patients and suggested that under the environment of limited medical resources, we should pay more attention to the ambulatory blood-pressure monitoring of patients with DKD.

Abnormal dipping status and nocturnal hypertension were found to be related to cardiovascular and renal outcomes in patients with CKD. Prospective observational studies showed that non-dipper BP patterns were relevant with renal outcomes, cardiovascular death and all-cause death events. On the contrary, severe clinical events could be avoided if these abnormal blood-pressure types were detected and managed at an early stage. In our study, the prevalence of abnormal dipper patterns such as reversed dipper pattern was higher in DKD patients than in PGD patients which indicated a higher risk for prognosis. In the same way, nocturnal hypertension was considered to be a risk factor for cardiovascular disease which was also confirmed by our study through the result of logistic regression; meanwhile, the prevalence of nocturnal hypertension in DKD patients was significantly higher than in PGD patients. Thus, the management of night-time BP and dipper pattern in type 2 DKD patients requires more focus. Considering that the abnormal dipping status and nocturnal hypertension can only be easily detected by ABPM, ABPM should be performed more in DKD patients than in PGD patients to recognize people with high cardiovascular risk.

Some research has explored the misclassification of BP pattern focusing on white-coat hypertension and masked hypertension in CKD patients. Patients with white-coat hypertension and masked hypertension showed higher risk for cardiovascular outcomes compared with people with normotension. In our study, DKD patients had a higher prevalence of sustained hypertension and lower prevalence of normotension compared with PGD patients. There was no significant difference in the prevalence of white-coat hypertension and masked hypertension for DKD and PGD patients. However, through stratified analysis by propensity-score weighting, DKD patients with white-coat hypertension, masked hypertension and sustained hypertension were more associated with target-organ damage than were PGD patients. Therefore, ABPM performed in DKD patients may be a better hint for TOD. 

A recent study has shown that abnormal blood-pressure patterns including non-dipping and reverse dipping blood-pressure pattern, masked hypertension and nocturnal hypertension detected by ambulatory blood-pressure monitoring in 150 normotensive diabetic patients were associated with concentric LVH and nephropathy. The common conclusion of our studies was that we both emphasized the importance of ABPM on patients with diabetes. The difference was that our study highlighted the contrast between DKD and PGD and pointed out the importance of DKD patients. In addition, our research objectives included not only normotensive patients but also patients with white-coat hypertension and sustained hypertension [38].

This study emphasized the different characteristics of ABPM between DKD patients and PGD patients and highlighted the importance of ABPM in DKD patients. All our patients were Asian and had comprehensive assessments, and all patients with dialysis were excluded in order to rule out the effect of hemodialysis on blood pressure. However, there were some limitations in our study. Firstly, we cannot infer a cause–effect relationship based on a cross-sectional study. Secondly, some information including the time of using antihypertensive drugs and the types of hypoglycemic drugs should be collected in detail. Thirdly, a single measurement of ambulatory blood pressure may be not enough. Finally, the median course of the disease was relatively short, but the mean GFR was 24 mL/min. We consider that this situation may be related to the fact that most patients do not usually have routine physical examinations. The subjects included in this study were all hospitalized patients. Therefore, most patients came to the hospital when they already had symptoms or signs related to renal injury or had found that their renal function was obviously impaired. It is difficult to accurately estimate the actual course of the disease, and the existing data can only be used as a reference. Therefore, there were some biases caused by population selection in this study. A larger sample size, multiple-center, prospective study is needed in the future.

## 10. Conclusions

Patients with type 2 diabetic kidney disease had more abnormal blood-pressure pattern were more closely related to target-organ damage than were patients with primary glomerular disease. Therefore, ambulatory blood-pressure monitoring should be performed in patients with type 2 diabetic kidney disease due to higher cardiovascular and renal risk.

## 11. Declarations

We would like to thank all of the patients and their families for participating in this study. This work was supported by the Five-five Project of the Fifth Affiliated Hospital of Sun Yat- sen University. The study protocol was approved by the ethics committee of our hospital (K14-1) and adhered to the Declaration of Helsinki. Written informed consent was given by all participants.

## Figures and Tables

**Figure 1 jcm-12-00167-f001:**
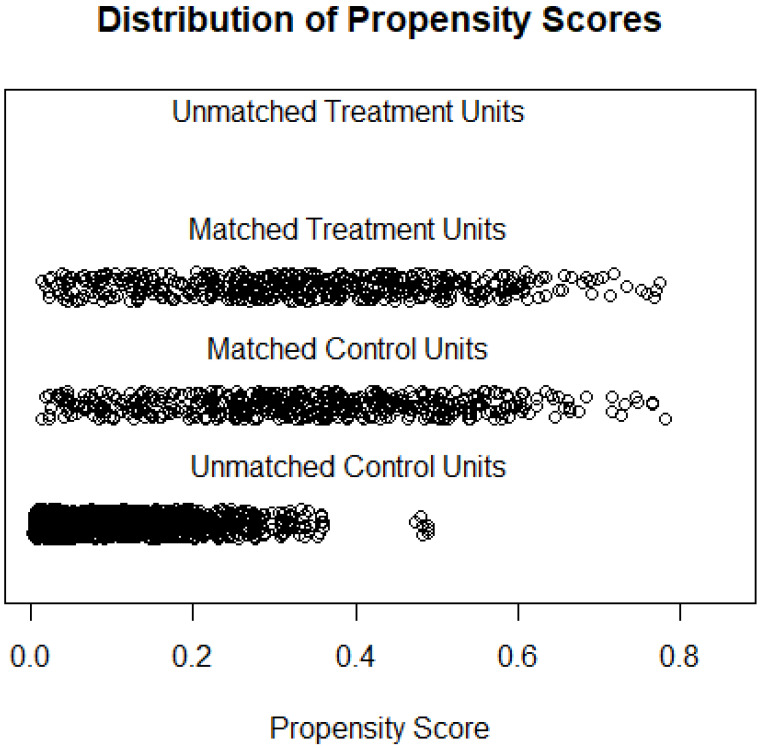
Distribution of propensity scores in the matching process.

**Figure 2 jcm-12-00167-f002:**
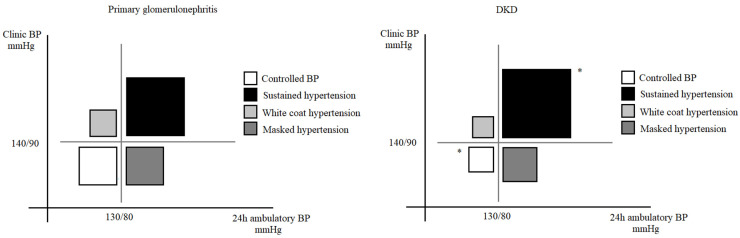
Proportions of different hypertension types through clinical and ambulatory blood pressure. Data are presented as numbers and proportions. Clinical hypertension—BP ≥ 140/90 mmHg. Ambulatory hypertension—BP ≥ 130/80 mmHg. White-coat hypertension—clinical HBP but normal ambulatory BP. Masked hypertension—normal clinical BP but ambulatory HBP. Normotension—normal clinical and ambulatory BP. Sustained hypertension—clinical and ambulatory HBP. DKD—diabetic kidney disease. PGD—primary glomerular disease. * indicates statistical difference between two groups.

**Figure 3 jcm-12-00167-f003:**
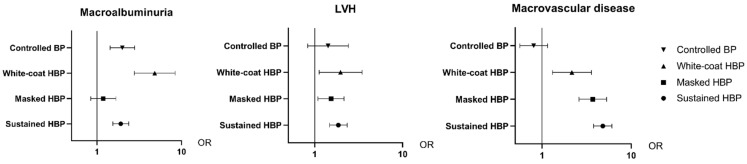
Stratified analysis for the association between different groups and target-organ damage in 4 different blood-pressure pattern groups using propensity score weighting (*n* = 1002). LVH—eft ventricular hypertrophy. OR—odds ratio.

**Table 1 jcm-12-00167-t001:** Demographic and clinical characteristics of type 2 DKD patients (*n* = 501) and PGD patients (*n* = 501).

Variable	Matched	
	PGD (*n* = 501)	DKD (*n* = 501)
Male (N/%)	323 (64.5%)	333 (66.5%)
Age (years)	57.9 ± 11.9	57.3 ± 10.5
BMI (kg/m^2^)	23.7 ± 3.5	24.8 ± 3.3 *
Current smoker (N/%)	165 (32.9%)	174 (34.7%)
Alcohol intake (N/%)	109 (21.8%)	125 (25.0%)
HBP family history (N/%)	35 (7.0%)	50 (10.0%)
Antihypertensive drug (N/%)	401 (80.0%)	442 (88.2%) *
RAS blockers (N/%)	170 (33.9%)	212 (42.3%) *
Statin (N/%)	112 (22.4%)	181 (36.1%) *
Hemoglobin (g/L)	110.8 ± 27.3	104.3 ± 26.7 *
Albumin (g/L)	36.1 ± 6.4	35.0 ± 6.4 *
Calcium (mg/dL)	8.5 ± 0.8	8.5 ± 0.9
Phosphate (mg/dL)	4.1 ± 1.4	4.5 ± 1.5 *
iPTH (pg/mL)	80.5 (46.9–190.7)	93.6 (67.9–124.0) *
HbA1c (%)	5.9 ± 1.1	7.0 ± 1.6 *
Serum fasting Glucose (mmol/L)	5.2 ± 1.5	7.0 ± 3.1 *
Cholesterol (mmol/L)	4.2 ± 2.3	5.0 ± 1.7 *
Triglyceride (mmol/L)	2.0 (1.2–3.7)	1.6 (1.1–2.4) *
HDL-C (mmol/L)	1.1 ± 0.4	1.1 ± 0.3
LDL-C (mmol/L)	3.1 ± 1.4	3.0 ± 1.2
Uric acid (mmol/L)	468.6 ± 132.8	466.5 ± 137.2
Serum creatinine (μmol/L)	224.0 (114.9–564.5)	305.8 (126.2–595.7)
eGFR (mL/min/1.73 m^2^)	24.0 (7.6–52.5)	17.3 (7.0–49.4)
UACR (mg/g)	441.4 (89.0–1410.2)	706.4 (174.2–3263.2) *
LVEF (%)	68.0 ± 8.1	65.7 ± 7.5 *
E/A	0.9 ± 0.3	0.9 ± 0.3
LVMI (g/m^2^)	112.0 ± 33.6	121.8 ± 28.6 *
Clinic-SBP (mmHg)	144.4 ± 22.8	153.2 ± 23.9 *
Clinic-DBP (mmHg)	85.5 ± 13.1	84.6 ± 13.1
24 h-SBP (mmHg)	133.9 ± 17.4	142.4 ± 17.8 *
24 h-DBP (mmHg)	82.7 ± 10.3	83.0 ± 9.3
Daytime-SBP (mmHg)	134.9 ± 17.3	143.2 ± 17.8 *
Daytime-DBP (mmHg)	83.6 ± 10.4	83.5 ± 9.3
Night time-SBP (mmHg)	129.5 ± 20.2	139.7 ± 20.6 *
Night time-DBP (mmHg)	78.9 ± 11.7	80.5 ± 11.2 *
Nocturnal hypertension (N/%)	420 (83.8%)	459 (91.6%) *

Data are presented as numbers (proportions), mean ± SD or median (interquartile range). * indicates statistical difference compared with PGD group, *p* < 0.05. DKD—diabetic kidney disease. PGD—primary glomerular disease. BMI—body mass index. HBP—hypertension. RAS blockers—renin-angiotensin system blockers. iPTH—intact parathyroid hormone. HbA1c—glycosylated hemoglobin, type A1C. HDL-C—high-density lipoprotein cholesterol. LDL-C—low-density lipoprotein cholesterol. eGFR—estimated glomerular filtration rate. UACR—urinary albumin to creatinine ratio. LVEF—left ventricular ejection fraction. SBP—systolic blood pressure. DBP—diastolic blood pressure.

**Table 2 jcm-12-00167-t002:** Prevalence of target-organ damage in patients with DKD (*n* = 501) and PGD (*n* = 501).

Variable	Matched	
	PGD (*n* = 501)	DKD (*n* = 501)
Macroalbuminuria (N/%)	301 (60.1%)	362 (72.3%) *
LVH (N/%)	236 (47.1%)	330 (65.9%) *
Macrovascular disease (N/%)	230 (45.9%)	316 (63.1%) *

Data are presented as numbers (proportions). * indicates statistic difference compared with the PGD group, *p* < 0.05. DKD—diabetic kidney disease. PGD—primary glomerular disease. LVH—left ventricular hypertrophy.

**Table 3 jcm-12-00167-t003:** Association between different groups, blood-pressure patterns and target-organ damage using logistic regression analysis. (*n* = 1002).

	Macroalbuminuria		LVH		Macrovascular Disease	
	OR (95%CI)	*p* Value	OR (95%CI)	*p* Value	OR (95%CI)	*p* Value
Model 1-DKD (vs. PGD)	1.730 (1.328–2.255)	<0.001	2.496 (1.771–3.426)	<0.001	2.139 (1.620–2.824)	<0.001
Model 2-DKD (vs. PGD)	1.707 (1.304–2.235)	<0.001	2.267 (1.715–2.999)	<0.001	2.107 (1.602–2.771)	<0.001
Model 3-DKD (vs. PGD)	1.483 (1.121–1.961)	0.006	2.167 (1.680–2.796)	<0.001	2.013 (1.563–2.591)	<0.001
Normal BP	Ref.		Ref.		Ref.	
White-coat HBP	3.134 (1.746–5.623)	<0.001	2.154 (1.184–3.917)	0.012	0.985 (0.540–1.797)	0.962
Masked HBP	2.173 (1.373–3.439)	0.001	2.919 (1.784–4.777)	<0.001	1.065 (0.654–1.733)	0.801
Sustained HBP	3.447 (2.305–5.157)	<0.001	3.576 (2.321–5.510)	<0.001	0.820 (0.537–1.251)	0.356

Data are presented as odds ratio (OR) and 95% confidence interval (95% CI). Model 1 was the univariate logistic regression of variate DKD (vs. PGD). Model 2 was the multivariate logistic regression of variate DKD (vs. PGD). Model 3 added classified variate BP pattern (vs. normotension) on the basis of Model 2. LVH—left ventricular hypertrophy. DKD—diabetic kidney disease. PGD—primary glomerular disease.

**Table 4 jcm-12-00167-t004:** Association between different groups, nocturnal hypertension (Yes/No) and target-organ damage using multivariate logistic regression analysis (*n* = 1002).

	Macroalbuminuria		LVH		Macrovascular Disease	
	OR (95%CI)	*p* Value	OR (95%CI)	*p* Value	OR (95%CI)	*p* Value
DKD (vs. PGD)	1.569 (1.187–2.074)	0.002	2.381 (1.809–3.136)	<0.001	2.086 (1.581–2.753)	<0.001
Nocturnal HBP	1.848 (1.233–2.771)	0.003	2.208 (1.459–3.343)	<0.001	1.668 (1.084–2.565)	0.020

Data are presented as odds ratio (OR) and 95% confidence interval (95% CI). LVH—left ventricular hypertrophy. DKD—diabetic kidney disease. PGD—primary glomerular disease.

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
