# Peer review of "Diabetic Kidney Disease versus Primary Glomerular Disease: A Propensity Score-Matched Analysis of Association between Ambulatory Blood-Pressure Monitoring and Target-Organ Damage"

_jcm, 2022, doi:10.3390/jcm12010167_

Round 1
Reviewer 1 Report
It is a huge amount of work done to evidence the difference in blood control between DKD and glomerular diseases.
The authors tried to make two comparable groups, but the main point that is not clearly shown is about the group of patients with primary glomerulonephritis. It was PGN with CKD GFR<60 ml.min.
There is no info about the treatment, the rate of remission the average time in CKD. I do not understand the principal value of the study. If the UACR ids are higher in DKD, it means that manly of the patients on PGN has been in some remission rate. The Higher value of proteinuria in DKD group and the higher value of UACR, and the low value of GFR ( although not statistically) were linked with a higher rate of macrovascular disease, which is expected.
The statistical analysis is good, and the tables are correct
The discussion needs a better analysis of the results The references need to be updated.
Author Response
Thank you very much for your comments on our paper. Your opinions and suggestions are very important and constructive. We have revised our paper according to your comments, and here’s our supplementary explanation:
- The median of the course of kidney disease was 12(2-48) months in two groups. The percentage of patients receiving antihypertensive therapy in DKD was 88.2% and was 80.0% in PGN group. 42.3% of DKD patients used RAS blockers and the number was 33.9% in PGN patients. In DKD group, 93.4% of the patients received hypoglycemic treatment and 42% of patients were treated with insulin.
- The DKD patients and PGD patients enrolled in this study were matched by the propensity score of age, sex and eGFR. Although the UACR was different in two groups, we included UACR as a variable in the statistical analysis to avoid data bias caused by UACR as much as possible. We found that the variate DKD(vs. PGD) was independently related to target organ damage, and we would like to emphasize the importance of patients with DKD through this study.
If you need any other information, please contact us. Thank you very much.
Yours Sincerely,
Dr. Yu
Dr. Wang

Reviewer 2 Report
The authors have conducted a rather original and well conceived study that may have useful clinical implications. The numerosity of the analised population is adequate, the statistical analysis appears well conducted and the paper has a clear structure.
Nevertheless in my opinion it is important to point out:
-the English language has to be revised e.g. lines 33-35; 48-58; 204; 282; 333
-some descriptions of abbreviations in tables and figures are missing
- I suggest to the authors to read this very recent paper ( Hemant Gupta, Tushar Vidhale, Manas Pustake, Charmi Gandhi & Tanmoy Roy (2022) Utility of ambulatory blood pressure monitoring in detection of masked hypertension and risk of hypertension mediated organ damage in normotensive patients with type 2 diabetes mellitus, Blood Pressure, 31:1, 50-57, DOI: 10.1080/08037051.2022.2061415) because I think it could be interesting and useful for your discussion and conclusions.
Author Response
Thank you very much for your comments on our paper. Over all the comments have been fair, encouraging and constructive. We have carefully proofread the manuscript according to your comments.
1.The English language has been revised according to your suggestion.
2.The descriptions of abbreviations has been added in tables and figures.
3.The paper you mentioned was very useful for our discussion and conclusions. In their study, abnormal blood pressure patterns including non-dipping and reverse dipping pattern, masked hypertension and nocturnal hypertension detected by ABPM in 150 normotensive diabetic patients were found to be associated with concentric LVH and nephropathy. In our study, we found that patients with DKD had more abnormal blood pressure pattern and had closer relationship with TOD than patients with PGD. The common conclusion of our studies was that we both emphasized the importance of ABPM on patients with diabetes. The difference was that our study highlighted the contrast between DKD and PGD, and pointed out the importance of DKD patients. Be-sides, our research object included not only normotensive patients but also patients with white-coat hypertension and sustained hypertension.
If you need any other information, please contact us. Thank you very much.
Very sincerely your,
Dr. Yu
Dr. Wang

Round 2
Reviewer 1 Report
The authors have improved the manuscript sufficiently.
In my opinion, one point is not yet clear concerning the PGN group. The data about this group are superficial. we know the median course of the disease, which is relatively short, described max 48 months. mean GFR was 24 ml/min. What are these primary glomerular diseases that, in a relatively short time, are associated with a critical GFR decline? Was that group in treatment apart from ACE-I that is prescribed additionally in the revised version?
Author Response
Thank you very much for your advice and guidance on our article.
The median course of the disease was relatively short, but the mean GFR was 24 ml/min. We consider that this situation may be related to the fact that most patients do not usually have routine physical examination. The subjects included in this study were all hospitalized patients. Therefore, most patients come to the hospital when they already have symptoms or signs related to renal injury or find that the renal function is obviously impaired. It is difficult to accurately estimate the actual course of the disease, and the existing data can only be used as a reference. Therefore, there were some biases caused by population selection in this study. A larger sample size, multiple-center, prospective study is needed in the future.
The treatment of the patients apart from ACE-I also included beta-blocker, calcium channel blocker, diuretic, α-receptor blocker and the number of antihypertensive drugs. Because the information is complex, it is difficult to collect all the accurate data, so this part of the data is not perfect for the time being. We will continue to work hard to improve this part of the data.
If you need any other information, please contact us. Thank you very much.
Yours Sincerely,
Dr. Yu
Dr. Wang